# Organocatalytic enantioselective $[2\pi + 2\sigma]$ cycloaddition reactions of bicyclo[1.1.0] butanes with $\alpha,\beta$-unsaturated aldehydes

Yi-Xiang Geng, Teng-Fei Xiao, Dong Xie, Ming-Ming Li, Pan-Pan Zhou, Guo-Qiang Xu ✉ & Peng-Fei Xu ✉

Bicyclo[2.1.1]hexanes (BCHs), three-dimensional benzene bioisosteres characterized by high sp³-carbon content, hold great promise for diverse applications in medicinal chemistry. Although significant advances have been made in the synthesis of racemic BCHs, highly enantioselective approaches remain comparatively rare. Here we report a mild, secondary amine–catalyzed asymmetric [2π + 2σ] cycloaddition of bicyclo[1.1.0]butanes (BCBs) with α,β-unsaturated aldehydes, which overcomes key limitations of existing metal-catalyzed and photochemical methods. The protocol operates under ambient air and tolerates a wide range of BCB and aldehyde substrates bearing diverse functional groups, affording BCH scaffolds in yields of up to 84% under Supramolecular Iminium Catalysis with excellent enantioselectivity (up to 99% ee) and high diastereoselectivity (>20:1 dr). The mild conditions and operational simplicity underscore the potential of this transformation for stereoselective manufacturing of BCHs at scale. Mechanistic experiments and DFT studies support an acid-promoted dual activation of both substrates, followed by an enamine–iminium tandem catalytic process that delivers the enantioenriched products.

Benzene rings represent the most frequently encountered structural motifs in pharmacopeia, serving either as primary bioactive cores or as topological anchors for pharmacophore alignment. Despite their ubiquity, recent studies have highlighted that the indiscriminate incorporation of phenyl rings often imparts suboptimal physicochemical properties to advanced molecules, ultimately limiting their development into effective therapeutic agents[1]. To overcome these drawbacks, the concept of "escaping from flatland" has emerged, emphasizing the design of F(sp³)-rich scaffolds. Increased carbon saturation enhances molecular solubility and three-dimensionality, thereby improving receptor binding selectivity, reducing off-target interactions, and raising the probability of clinical success[2,3]. For example, Stepan and Mykhailiuk independently demonstrated that substituting phenyl rings with saturated three-dimensional bioisosteres can significantly improve drug-like properties[4-7]. Among such bioisosteres, bicyclo[2.1.1]hexanes (BCHs) have emerged as privileged replacements for aromatic rings (Fig. 1a). They offer unique opportunities to simultaneously optimize pharmacological performance and expand intellectual property space, which has spurred strong interest in developing efficient synthetic routes[8-12]. One of the most powerful strategies relies on the [2π + 2σ] cycloaddition of bicyclo[1.1.0]butanes (BCBs) with π-systems, a transformation that delivers BCHs with excellent atom economy and synthetic efficiency[13-17].

Significant progress has been made toward the synthesis of racemic BCHs, notably via Lewis acid catalysis[18-29], radical-mediated strategies[30-38], and triplet-energy-transfer-enabled

State Key Laboratory of Nature Product Chemistry, College of Chemistry and Chemical Engineering, Lanzhou University, Lanzhou, PR China.
✉ e-mail: gqxu@lzu.edu.cn; xupf@lzu.edu.cn

**Fig. 1 | Synthetic approaches to BCHs and key strategies for enantioselective BCHs synthesis. a** BCHs as 3D benzene bioisosteres. **b** Previous works focused on construction of BCHs. **c** Previous work: Enantioselective [2π + 2σ] cycloaddition via metal or photocatalysis. **d** This work: Organocatalytic enantioselective [2π + 2σ] cycloaddition.

[2π + 2σ] cycloadditions (Fig. 1b)[39–42]. However, studies by Baran and co-workers have shown that enantiomerically distinct BCH analogs can exhibit dramatically different biological activities when serving as phenyl bioisosteres[43]. This underscores the critical importance of developing stereoselective methods for BCH synthesis. Early contributions include Bach's pioneering work, which achieved enantioselective additions of BCBs to quinolones using a chiral template (two equivalents) combined with triplet-energy transfer under 366 nm irradiation at −60 °C[44]. Subsequently, Jiang's group reported a bifunctional chiral phosphoric acid–catalyzed process between BCBs and vinylazaarenes under blue LED light at −40 °C[45]. More recently, Hong and Liu independently realized asymmetric [2π + 2σ] cycloadditions of BCBs with α,β-unsaturated ketones and coumarins using Lewis acid catalysis with chiral ligands (Fig. 1c)[46,47]. In parallel, Zheng and Zi independently reported palladium-catalyzed enantioselective additions of vinyl-carbonyl-BCBs to alkenes[48,49]. Photocatalytic asymmetric cycloadditions of BCBs with α,β-unsaturated ketones, vinyl azides, and naphthalenes have also been disclosed by Yoon, Zheng, and You using chiral Lewis or Brønsted acid systems (see Supplementary Fig. S1)[50–52].

While these elegant studies provide viable strategies for accessing enantioenriched BCHs, they generally suffer from several drawbacks: (i) reliance on cryogenic conditions under inert atmospheres, (ii) requirement for specialized photochemical setups, and (iii) use of potentially toxic transition metals, which diminishes their suitability for pharmaceutical applications. To overcome these limitations, we envisioned a mechanistically distinct approach–leveraging secondary amine catalysis under ambient air, at room temperature, and under operationally simple conditions. This strategy not only expands the synthetic toolbox for BCHs but also addresses key challenges in developing practical, sustainable, and pharmaceutically compatible methods.

## Results

### Condition optimization

To evaluate the feasibility of our design, we first examined the reaction between cinnamaldehyde (**1a**) and pyrazole amide–substituted bicyclo[1.1.0]butane (**2a**) as model substrates (Table 1). A range of secondary amines were screened. With imidazolinone **4a** as the catalyst, the desired product **3a** was obtained in 75% yield but with only 8% ee. Seeking improved enantioselectivity, we tested sterically hindered catalyst **4b**; however, the outcome was unsatisfactory. Seeking improved enantioselectivity, we tested sterically hindered catalyst **4b**; however, the outcome was unsatisfactory. In contrast, the use of diarylprolinol silyl ether **4c** enhanced both yield and enantioselectivity. Encouraged by this improvement, we further tested the more reactive and sterically demanding catalyst **4d**. Although the yield decreased slightly, product **3a** was obtained with very high enantioselectivity. We next screened solvents and found acetone to be the optimal medium. To further enhance efficiency, we incorporated hydrogen-bonding catalyst **5** based on our previously established supramolecular iminium ion catalysis concept, followed by comprehensive screening and optimization (see Supplementary Table S1)[53–56]. Ultimately, the use of auxiliary catalyst **5a** afforded **3a** in 82% yield and 99% ee within 24 h.

### Substrate scope

With optimal conditions established, we explored the substrate scope of aldehydes (Fig. 2). Substituents such as methyl (**3b**), methoxy (**3c**), or halides (**3d**–**3f**) at the para-position of cinnamaldehyde maintained excellent enantioselectivity, though yields were slightly reduced. Similar trends were observed for meta-substitution (**3m**–**3r**), highlighting sensitivity to electronic effects. Aldehydes bearing a bulky tert-butyl group (**3i**) also gave excellent enantioselectivity. Importantly, substrates featuring synthetically versatile groups–including nitro (**3g**), cyano (**3h**), ester (**3j**), trifluoromethylthio, and biphenyl moieties (**3k, 3l**)–afforded the

**Table 1 | Optimization of the reaction conditions[a]**

| Entry | Catalyst | Solvent | Yield[b](%) | ee[c](%) |
|---|---|---|---|---|
| 1 | 4a | Acetone | 75 | 8 |
| 2 | 4b | Acetone | 48 | 22 |
| 3 | 4c | Acetone | 92 | 89 |
| 4 | 4d | Acetone | 85(72[d]) | 99 |
| 5 | 4d | DCM | 13 | n.d. |
| 6 | 4d | MeCN | 53 | 99 |
| 7 | 4d | THF | trace | n.d. |
| 8 | 4d | EA | trace | n.d. |
| 9 | 4d | Toluene | trace | n.d. |
| 10[e] | 4d | Acetone | 82[d] | 99 |

n.d. not determined, TBS tert-butyldi-methylsilyl, TDS thexyl-dimethylsilyl, TFA trifluoroacetic acid.
[a]Reaction conditions: Reactions performed with **1a** (0.1 mmol), **2a** (0.15 mmol), catalyst (20 mol%), TFA (40 mol%) in solvent (2 mL) at 10 °C for 36 h.
[b]Determined through gas chromatography (GC) analysis.
[c]Determined through high-performance liquid chromatography (HPLC) analysis.
[d]Yield of isolated product.
[e]With auxiliary **cat. 5a** (20 mol%).

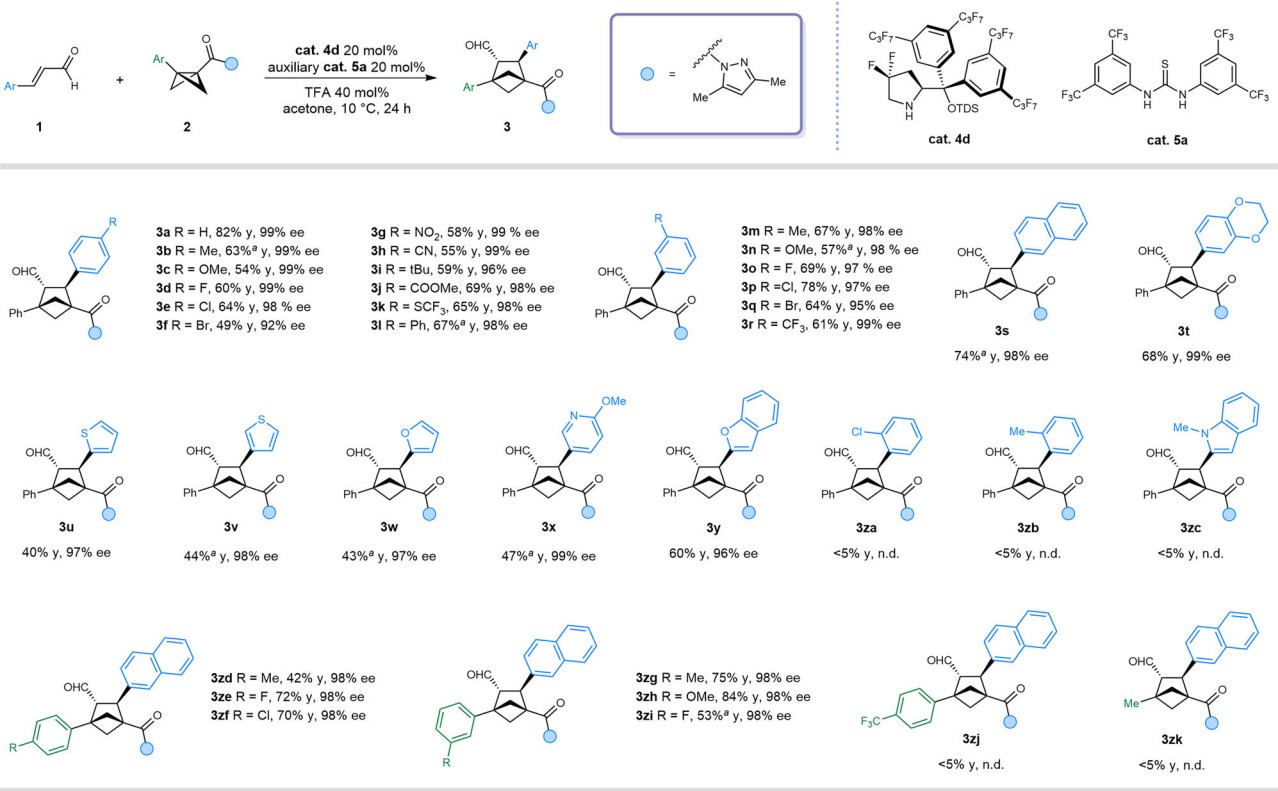

**Fig. 2 | Scope of substrates.** Standard reaction conditions: aldehyde (0.1 mmol), BCB (0.15 mmol), **cat. 4 d** (20 mol%), **cat. 5a** (20 mol%), TFA (40 mol%) and acetone (2 mL).

desired products in moderate yields with consistently high enantioselectivity. Beyond substituted phenyl groups, fused aryl and heteroaryl aldehydes proved compatible, delivering products **3s–3y** with excellent enantioselectivity. However, ortho-substituted substrates (**3za–3zc**) suffered from steric hindrance that impeded reactivity. Interestingly, employing a naphthyl-substituted aldehyde not only improved reaction yield but also facilitated product isolation. Thus, substrate **1s** was selected as the representative aldehyde for exploring the scope of BCBs. Substituted BCBs exhibited varied reactivity. Halogen or electron-donating groups at the para- or meta-positions of the phenyl ring afforded good results (**3ze–3zi**). However, para-electron-donating substituents led to competitive self-ring-opening under acidic conditions, reducing the yield to 42% (**3zd**). In contrast, strongly electron-withdrawing groups such as para-trifluoromethyl suppressed BCB activation, producing only trace product (**3zj**). Furthermore, BCBs bearing groups other than acyl pyrazole proved unstable under the reaction conditions, leading to low yields and facile decomposition[57]. By comparison, the acyl-pyrazole–substituted BCB exhibited superior stability, underscoring its unique suitability for this transformation (see Supplementary Fig. S4 for unsuccessful substrates).

### Product derivatization
To assess the practicality of this protocol, we carried out a gram-scale synthesis of product **3s** (3 mmol). The reaction maintained excellent enantioselectivity, with only a slight decrease in yield (−4%). The exposed aldehyde group of **3s** proved highly versatile for downstream transformations. Under NaH conditions, it underwent a Wittig reaction to afford alkene **6** in 65% yield with complete retention of enantioselectivity. Reduction with NaBH₄ produced alcohol **7** in 86% yield and 98% ee. The amide–pyrazole group could be smoothly converted into a methyl ester in 99%

yield under DBU/MeOH conditions, while aldehyde protection with ethylene glycol afforded acetal **9**, enabling single-crystal X-ray analysis (CCDC 2441435). Moreover, the aldehyde functionality was transformed into a terminal alkyne in 96% yield using the Bestmann–Ohira reagent. This alkyne intermediate served as a versatile handle for further diversification: it participated in a Yamanaka–Sakamoto–Sonogashira indole synthesis to furnish indole **11** (92% yield, 94% ee), underwent Sonogashira coupling with aryl iodides to provide **12** (79% yield, 93% ee), and engaged in CuAAC "click" chemistry to yield triazole **13** (85% yield, 91% ee) (Fig. 3).

We further demonstrated the utility of **3s** by oxidizing the aldehyde group to carboxylic acid **14** via Pinnick oxidation (Fig. 4). This transformation greatly expanded opportunities for late-stage functionalization with bioactive molecules. Through amidation or esterification, we successfully conjugated **14** with amine-containing compounds–including tryptamine (**16**), rivaroxaban (**17**), fluvoxamine (**18**), oseltamivir (**20**), dehydroabietylamine (**21**), and linagliptin (**25**)–as well as alcohol-containing molecules such as geraniol (**15**), cholesterol (**19**), diacetone-D-glucose (**22**), pregnenolone (**23**), L-menthol (**24**), and testosterone (**26**). These reactions furnished a diverse library of biologically relevant BCH analogs in good to excellent yields (58–98%).

### Mechanistic investigation
To gain insight into the reaction pathway, we performed a series of control experiments (Fig. 5a). The reaction did not proceed in the absence of either the secondary amine catalyst or the acid, demonstrating that both components are essential. To clarify the role of the acid, we systematically evaluated a range of acids. Neither weakly acidic benzoic acid (BA) nor strongly acidic p-toluenesulfonic acid (TsOH) delivered the target product (Fig. 5b). ¹H NMR studies

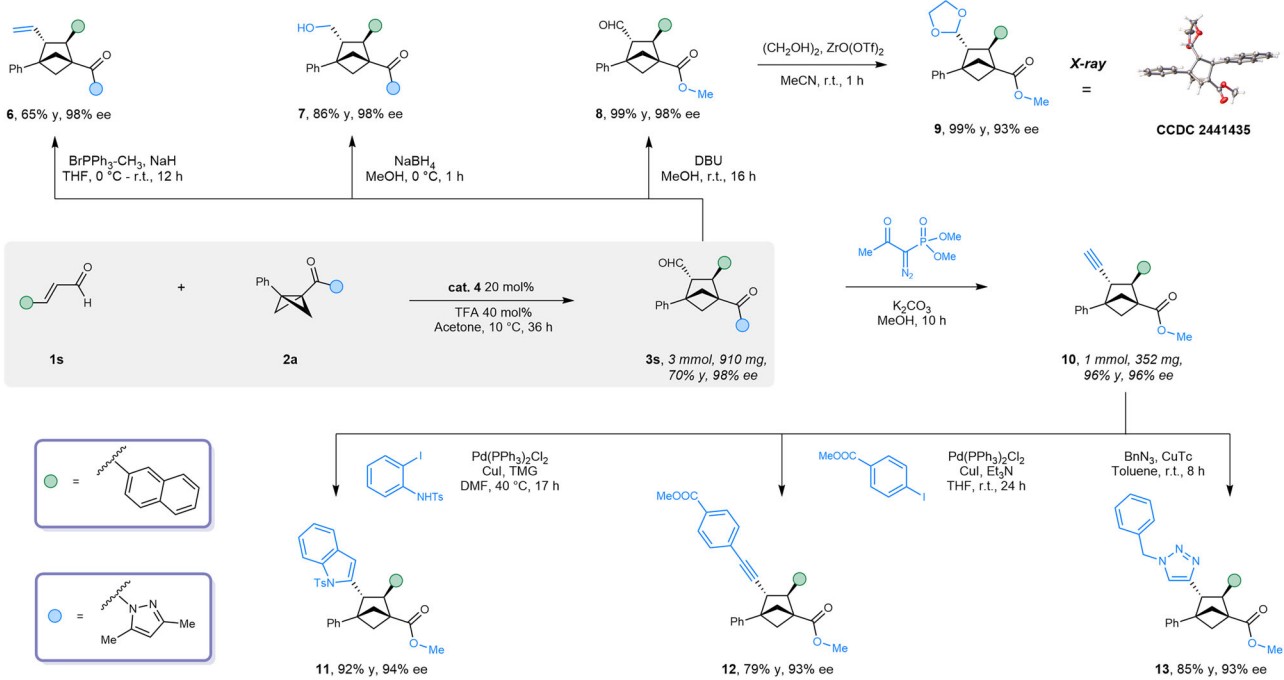

**Fig. 3 | Applications.** Scale-up synthesis and post-synthetic modifications.

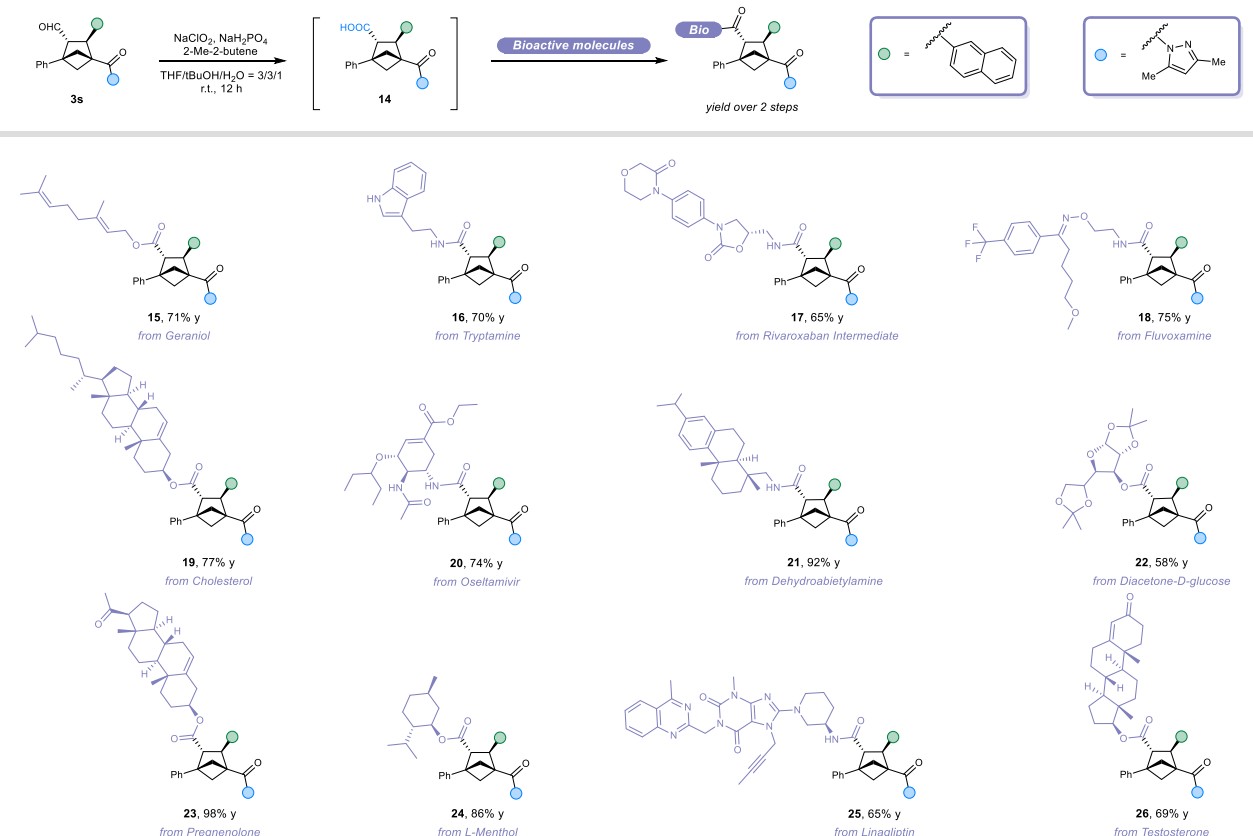

**Fig. 4 | Late-stage modification of drugs.** Late-stage functionalization of the BCH scaffold for bioactive molecule diversification.

(Supplementary Fig. S5) confirmed that BA failed to promote condensation between the aldehyde and secondary amine, thereby preventing iminium ion formation and halting the reaction. In contrast, TsOH successfully promoted iminium ion generation but its excessive acidity induced rapid ring-opening of the BCB substrate, leading to nearly quantitative formation of ring-opened byproducts and preventing product formation. When **1a** was treated with acid alone, we isolated product **27**, confirming that BCB ring-opening can occur under acidic conditions in the absence of the amine catalyst.

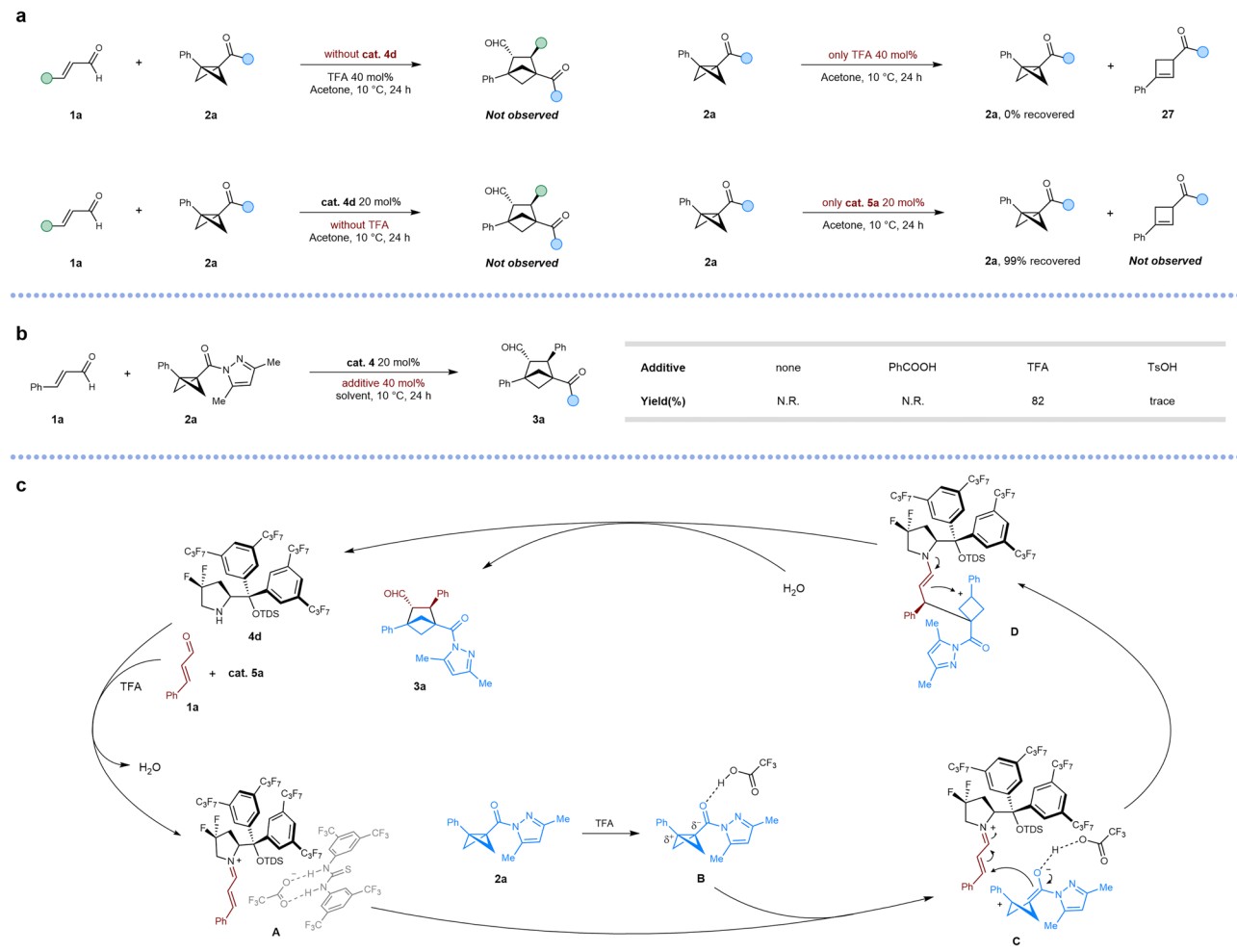

**Fig. 5 | Proposed mechanism. a** Control experiments. **b** Reaction with different acid. **c** Proposed catalytic cycle.

Based on these results, we propose the mechanism outlined in Fig. 5c. The secondary amine catalyst **4d** and hydrogen-bonding catalyst **5a** undergo acid-promoted condensation with aldehyde **1a** to form supramolecular iminium intermediate **A**. The acid-activated BCB then attacks the methylene carbon of **A**, generating carbocation intermediate **D**, which subsequently undergoes cyclization with the enamine. The tandem enamine–imine catalytic cycle ultimately regenerates the secondary amine catalyst, affording the target product **3**.

### DFT calculations

To further support this proposal, density functional theory (DFT) calculations were carried out (Fig. 6). The transition state **ts-a-II**, formed from **2a** in the presence of TFA, was stabilized by 5.6 kcal/mol compared with the TFA-free system **ts-a**, with subsequent steps also exhibiting reduced energy barriers. These results corroborate the role of TFA in activating the BCB moiety. Additionally, we evaluated the effect of catalyst **5a** on the iminium ion pair. Prior to its introduction, the N–O distance measured 2.92 Å. Incorporation of **5a** led to a 1.6 kcal/mol stabilization and increased the N–O distance to 3.20 Å. This indicates that **5a** weakens electrostatic interactions between the trifluoroacetate anion and the iminium cation, thereby stabilizing the ion pair and improving reactivity. Notably, catalyst **5a** also enhances the electrophilicity of the iminium double bond, particularly at the methylene carbon, facilitating nucleophilic attack by the BCB-derived carbanion. These computational insights strongly support the cooperative catalytic role of **5a** in the reaction.

### Discussion

In summary, we have developed a secondary amine–catalyzed enantioselective [2π + 2σ] cycloaddition of BCBs with α,β-unsaturated aldehydes. The reaction proceeds efficiently under ambient conditions without the need for an inert atmosphere, displaying broad functional group tolerance and delivering products with outstanding enantioselectivity (up to 99% ee). Beyond the direct transformation, the resulting products proved highly versatile, undergoing a variety of downstream modifications to furnish structurally and functionally diverse derivatives. Mechanistic investigations, supported by control experiments and DFT studies, confirmed an acid-promoted secondary amine catalytic pathway. Overall, this work establishes a robust and practical platform for accessing chiral BCH scaffolds and offers a valuable complement to the limited existing methods for asymmetric BCB cycloadditions.

### Methods

#### General procedure for enantioselective cycloaddition of and α,β-unsaturated aldehydes and bicyclo[1.1.0]butanes

To a 10 mL Schlenk tube equipped with a magnetic stir bar was added cat. **4d** (0.02 mmol, 0.2 eq.), cat. **5a** (0.02 mmol, 0.2 eq.), aldehydes **1** (0.1 mmol, 1.0 eq.), BCBs **2** (0.15 mmol, 1.5 eq.) and TFA (0.04 mmol, 0.4 eq.), then acetone (2 mL) was added. The resulting mixture was stirred at 10 °C for 24 h. Upon completion of the reaction, the reaction mixture was concentrated under reduced pressure, and the resulting crude mixture was purified by silica gel column chromatography to afford the pure product **3**.

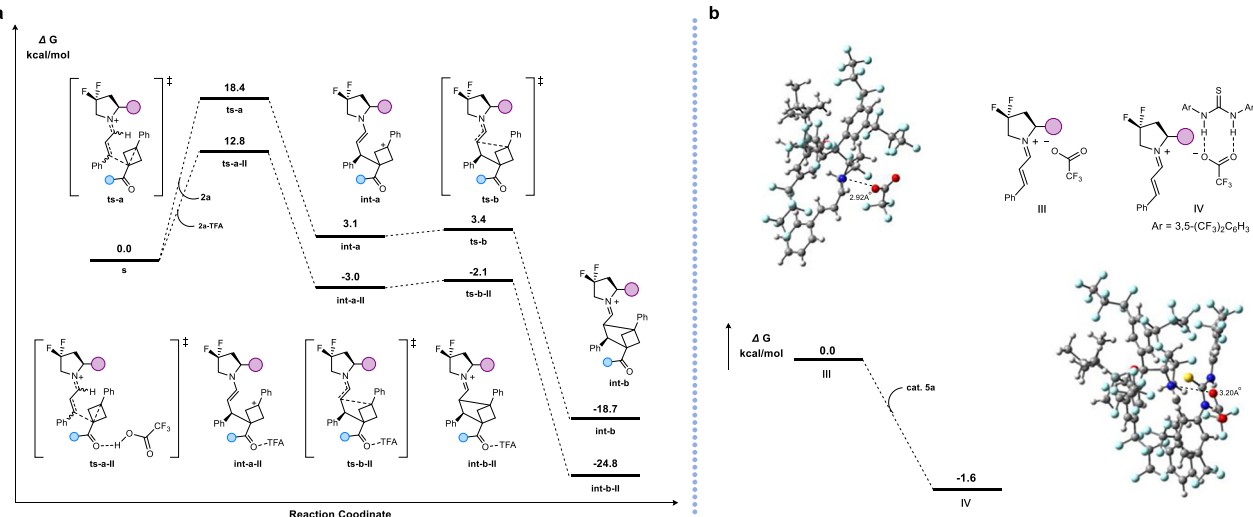

**Fig. 6 | DFT calculations. a** Investigated whether TFA assists in the ring-opening of BCBs through DFT calculations. **b** Investigated the influence of hydrogen-bonding catalysts on iminium ion pairs through DFT calculations.

## Data availability
The data that support the findings of this study are available within the main text and its Supplementary Information. Crystallographic data for the structures reported in this Article have been deposited at the Cambridge Crystallographic Data Centre, under deposition numbers CCDC 2441435 (9). Copies of the data can be obtained free of charge via https://www.ccdc.cam.ac.uk/structures/. All data are available from the corresponding author upon request. Source data are provided with this paper.

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

## Acknowledgements

We are grateful to the NSFC (U22A20390 and 22371098), supported by the Xinjiang Uygur Autonomous Region Science and Technology Department's project (No. Xincaihang [2023-211]), the Science and Technology Major Program of Gansu Province of China (22ZD6FA006, 23ZDFA015, 23JRRA1512) and the "111" program from the MOE of PR China.

## Author contributions

Y.-X.G., T.-F.X. and M.-M.L. carried out the experiments and data analysis work. P.-P.Z. and D.X. performed the DFT calculations. G.-Q.X. and P.-F.X. designed the reaction and directed the project. The paper was written by G.-Q.X. and P.-F.X. All authors contributed to discussions.

## Competing interests

The authors declare no competing interests.
