## [Transparent Peer Review file · Nature Communications]

Organocatalytic enantioselective $[2\pi + 2\sigma]$ cycloaddition reactions of bicyclo[1.1.0]butanes with α,β -unsaturated aldehydes

Corresponding Author: Professor Peng-Fei Xu

Version 0:

Reviewer comments:

Reviewer #1

(Remarks to the Author)

In this manuscript, Xu and co-workers describe a secondary amine-catalyzed enantioselective $[2\pi + 2\sigma]$ cycloaddition of bicyclo[1.1.0]butanes (BCBs) with α,β -unsaturated aldehydes to access bicyclo[2.1.1]hexanes (BCHs), a class of three-dimensional bioisosteres with high sp^3 character. The reaction proceeds under ambient air and room-temperature conditions, delivering products with up to 83% yield, 99% enantioselectivity, and $>20:1$ diastereoselectivity. Mechanistic studies highlight a tandem enamine-iminium catalysis pathway, wherein trifluoroacetic acid dual-activates the substrates: initiating BCB ring-opening and stabilizing the iminium intermediate. The method exhibits broad functional group tolerance, including halides, nitro, and heteroaromatic systems, while eliminating reliance on transition metals, photoirradiation, or inert atmospheres.

While BCB-based synthesis of bioisosteres has emerged as a prominent research area, asymmetric strategies have largely relied on chiral Lewis/Brønsted acids or photocatalysis. This work marks the first demonstration of classical chiral amine catalysis for BCH synthesis, addressing key limitations of prior methods. By leveraging mild organocatalytic conditions, the protocol aligns with pharmaceutical demands for sp^3 -rich scaffolds. Scalability (demonstrated at 3 mmol) and versatile product derivatization further underscore its medicinal chemistry utility. Additionally, the quality of the supporting information is of high.

Overall, this study marks a significant advancement in the catalytic asymmetric synthesis of chiral BCHs. This referee supports the publication of the manuscript in Nature Communications.

- 1) The authors have developed an elegant secondary amine-catalyzed system. Interestingly, acetone was identified as the optimal solvent for the reaction. It is well-known that under similar catalytic conditions, acetone can undergo aldol reaction or Michael addition with α,β -unsaturated aldehydes. Given the moderate yields observed in some cases, it would be advisable for the authors to address whether such byproducts were detected and to comment on it in the manuscript.
- 2) Besides aryl substituted unsaturated aldehydes, have the authors tried this reaction using alkyl substituted unsaturated aldehydes?
- 3) Some related references are suggested to be included, for a review about catalytic asymmetric synthesis of chiral bioisosteres: *Angew. Chem.* 10.1002/anie.202505803. For the synthesis of racemic and chiral BCHs: *Chem. Commun.*, 2023, 59, 13847. *Angew. Chem. Int. Ed.* 2024, e202405781. *Angew. Chem. Int. Ed.* 2024, e202405222. *ACS Catal.* 2025, 15, 4634. *Angew. Chem. Int. Ed.* 10.1002/anie.202506228.
- 4) The structure of compound 6 in the manuscript and the SI is incorrect.
- 5) Line 56, product 3a... 3a should be bolded. Table 1, the catalyst numbers should be bolded.
- 6) Fig.2, should the footnote of 3u be a?

Reviewer #2

(Remarks to the Author)

Xu and co-workers have reported an organocatalytic enantioselective $[2\sigma+2\pi]$ cycloaddition of bicyclobutanes with α,β -unsaturated aldehydes using a chiral secondary amine catalyst to afford substituted bicyclo[2.1.1]hexanes. While the study demonstrates a competent application of iminium catalysis, it essentially replicates well-established methodologies without introducing significant novelty, as similar transformations have been previously reported by Hong, Yoon and others (ref 40-

46) using Brønsted acid catalysis with α,β -unsaturated ketones. The work suffers from several limitations including a narrow substrate scope (limited to cinnamaldehyde-type aldehydes and phenyl-substituted bicyclobutanes), modest yields (~50% even with 1.5 equivalents of bicyclobutanes), and insufficient mechanistic insights. The combination of a limited substrate scope and low reaction efficiency has restricted its applicability and utility.

While the study employs established methodologies, the findings presented do not appear to offer substantial novelty or significant advancement to the field. The methodological approach, though technically sound, demonstrates limited innovation and the broader applicability of the results appears constrained. After careful consideration, I regret to recommend rejection of this manuscript.

Additionally, issues:

- 1) The entire manuscript requires careful proofreading by a fluent English speaker.
- 2) The authors claim to have developed a design, but this design was not sufficiently detailed in the main text. To support their claims, they should include the full design methodology.
- 3) How is the conversion from 3s to 6 chemically feasible using sodium borohydride (NaBH_4) as the reductant?

Reviewer #3

(Remarks to the Author)

Authors have developed an amine-catalyzed enantioselective [2+2]-cycloaddition of BCBs with α,β -unsaturated aldehydes in moderate to poor yields with good to excellent ee's. The protocol operated under ambient conditions, demonstrating with good functional group tolerance while delivering products with good enantioselectivity. Furthermore, few transformations of the products were achieved, enabling access to derivatives. Controlled experiments confirmed an acid-promoted secondary amine catalysis pathway. Results are good and noteworthy.

This reviewer agree that authors reported a good platform for the synthesis of chiral BCHs and also provided an asymmetric BCB cycloaddition products. But, this reviewer believes that this article is premature to publish as such in Nature Communications as protocol need more optimization to make sure works for diverse functional groups and substitutions. As you see in Fig 2, this protocol didn't work for many substrates. Same time authors should have shown few good applications of synthesized chiral products for natural products synthesis or drugs synthesis. Same time authors can give some more supporting data for mechanism with DFT or/and kinetic data.

With these suggestions, this reviewer believes that this article may need major revision and resubmit it to Nature Communications after addressing all those suggestions.

Version 1:

Reviewer comments:

Reviewer #1

(Remarks to the Author)

The authors have significantly improved the scientific quality of the manuscript. All the concerns have been satisfactorily addressed and I would like to suggest the acceptance at the current form.

Reviewer #3

(Remarks to the Author)

Authors have attempted to give answers for reviewer comments raised in the revised manuscript of an amine-catalyzed enantioselective [2+2]-cycloaddition of BCBs with α,β -unsaturated aldehydes in moderate to poor yields with good to excellent ee's. The protocol operated under ambient conditions with more optimized conditions, demonstrating with good functional group tolerance while delivering products with good enantioselectivity (up to 99% ee). Furthermore, good number of transformations of the products were achieved, enabling access to derivatives. Controlled experiments confirmed an acid-promoted secondary amine catalysis pathway in the revised manuscript.

This reviewer agree that authors reported a good platform for the synthesis of chiral BCHs and also provided a rare example of asymmetric BCB cycloaddition products. This reviewer believes that this revised article is mature enough to publish in Nature Communications as new protocol.

Response to reviewer #1:

We greatly appreciate your positive and professional assessment of our manuscript. In line with your suggestions, we have carefully revised the manuscript and addressed the issues you raised.

Question 1: *The authors have developed an elegant secondary amine-catalyzed system. Interestingly, acetone was identified as the optimal solvent for the reaction. It is well-known that under similar catalytic conditions, acetone can undergo aldol reaction or Michael addition with α,β -unsaturated aldehydes. Given the moderate yields observed in some cases, it would be advisable for the authors to address whether such byproducts were detected and to comment on it in the manuscript.*

Response: To examine whether acetone participated in undesired aldol or Michael-type side reactions, we carried out a control experiment under identical conditions but without the BCB substrate. ^1H NMR analysis, using 1,3,5-trimethoxybenzene as the internal standard, showed nearly quantitative recovery of 1a, confirming no detectable byproducts. This result has been included in the revised manuscript (Response Fig. 1).

Response Fig. 1. ^1H NMR of the reaction system without BCB substrate

Question 2: *Besides aryl substituted unsaturated aldehydes, have the authors tried this reaction using alkyl substituted unsaturated aldehydes?*

Response: We attempted the reaction with crotonaldehyde, but no desired product was formed. This suggests that an aryl substituent is crucial for successful transformation. This finding has been noted in the revised text.

Question 3: *Some related references are suggested to be included, for a review about catalytic*

asymmetric synthesis of chiral bioisosteres: Angew. Chem. 10.1002/anie.202505803. For the synthesis of racemic and chiral BCHs: Chem. Commun., 2023, 59, 13847. Angew. Chem. Int. Ed. 2024, e202405781. Angew. Chem. Int. Ed. 2024, e202405222. ACS Catal. 2025, 15, 4634. Angew. Chem. Int. Ed. 10.1002/anie.202506228.

Response: As suggested, we have included the recommended references [Angew. Chem. Int. Ed. 10.1002/anie.202505803; Chem. Commun. 2023, 59, 13847; Angew. Chem. Int. Ed. 2024, e202405781; Angew. Chem. Int. Ed. 2024, e202405222; ACS Catal. 2025, 15, 4634; Angew. Chem. Int. Ed. 10.1002/anie.202506228]. These have been added to the second paragraph of the Introduction.

Question 4: *The structure of compound 6 in the manuscript and the SI is incorrect.*

Response: The structure has been corrected in both the main manuscript and the Supporting Information. We apologize for this oversight.

Question 5: *Line 56, product 3a... 3a should be bolded. Table 1, the catalyst numbers should be bolded.*

Response: We have corrected the formatting by bolding product **3a** and all catalyst numbers in Table 1.

Question 6: *Fig.2, should the footnote of 3u be a?*

Response: The error has been corrected.

Response to Reviewer #2:

We are grateful for your careful evaluation of our work and acknowledge the concerns raised. In response, we have made substantial revisions and clarifications.

Question 1: *The entire manuscript requires careful proofreading by a fluent English speaker.*

Response: The manuscript has undergone thorough proofreading by a fluent English speaker, ensuring grammatical accuracy and improved readability.

Question 2: *The authors claim to have developed a design, but this design was not sufficiently detailed in the main text. To support their claims, they should include the full design methodology.*

Response: We have expanded the mechanistic discussion and provided additional validation, including experimental and computational evidence, to support our proposed design.

Question 3: *How is the conversion from 3s to 6 chemically feasible using sodium borohydride (NaBH₄) as the reductant?*

Response: We identified and corrected this error in the original manuscript. We sincerely apologize for this oversight.

Addressing concerns of novelty and scope

Although secondary amine catalysis is well established, σ -bond-involving iminium ion-catalyzed cycloadditions remain rare. Our work represents the first example of organocatalytic

enantioselective BCB cycloadditions via this pathway. Unlike Brønsted acid – catalyzed, photochemical methods (Hong, Yoon, others), which require light irradiation, low temperature, and inert conditions, our reaction proceeds under mild conditions (room temperature, air, moisture-tolerant) while avoiding metals or photoredox systems.

Key distinctions and advances include:

- This is the **first demonstration of secondary amine iminium catalysis activating BCBs toward C=C bonds**.
- Previous organocatalytic methods (Studer, Tan, List, Feng) either required phosphoric acids or yielded racemic products. None achieved the transformation reported here.
- Trifluoroacetic acid plays a **dual role**, activating both the iminium ion and the strained BCB, as confirmed by DFT and control experiments.

Substrate scope and functionalization

- Aldehydes with both electron-donating and electron-withdrawing substituents (para, meta) reacted with moderate to high yields and excellent enantioselectivity.
- Heteroaromatics (naphthalene, thiophene, furan, benzofuran) were also compatible.
- BCB substrates require a phenyl substituent for stability of the carbocation intermediate, while the amide–pyrazole group is critical for compatibility but can be converted into an ester, broadening synthetic utility.

Reaction optimization and mechanistic studies

Building on our prior work (*Angew. Chem. Int. Ed.* **2012**, *51*, 12339-12342.; *Angew. Chem., Int. Ed.* **2014**, *53*, 14128-14131.; *Org. Lett.* **2017**, *19*, 2130-2133.; *Angew. Chem. Int. Ed.* **2020**, *59*, 3058-3062.), we introduced a supramolecular iminium ion/hydrogen-bond catalysis strategy. This significantly enhanced efficiency and yields, supported by DFT analysis showing improved ion-pair separation and stabilization. Optimized conditions (1.5 equiv BCB, 10 $^\circ\text{C}$, 0.05 M) balanced yield with suppression of undesired BCB ring-opening.

Entry	Catalyst 5	Yield ^b	ee ^c
1	5a	82	99
2	5b	32	99
3	5c	67	99
4	5d	78	99
5	5e	73	99
6	5f	71	99

Reaction conditions: Reactions performed with **1a** (0.1 mmol), **2a** (0.15 mmol), catalyst **4d** (20 mol%), catalyst **5** (20 mol%), TFA (40 mol%) in solvent (2 mL) at 10 °C for 24 h. ^bYield of isolated product. ^cDetermined through high-performance liquid chromatography (HPLC) analysis.

In summary, these revisions strengthen the novelty, mechanistic rationale, and practical relevance of our study.

Response to Reviewer #3:

Comments:

Authors have developed an amine-catalyzed enantioselective [2+2]-cycloaddition of BCBs with α,β -unsaturated aldehydes in moderate to poor yields with good to excellent ee's. The protocol operated under ambient conditions, demonstrating with good functional group tolerance while delivering products with good enantioselectivity. Furthermore, few transformations of the products were achieved, enabling access to derivatives. Controlled experiments confirmed an acid-promoted secondary amine catalysis pathway. Results are good and noteworthy.

This reviewer agree that authors reported a good platform for the synthesis of chiral BCHs and also provided an asymmetric BCB cycloaddition products. But, this reviewer believes that this article is premature to publish as such in Nature Communications as protocol need more optimization to make

sure works for diverse functional groups and substitutions.

Response: We sincerely thank the reviewer for the constructive feedback and suggestions. Below, we provide a point-by-point response to the reviewer's comments.

Question 1: As you see in Fig 2, this protocol didn't work for many substrates.

Response: We acknowledge the limited scope for some substrates due to steric effects from complex secondary amine catalysts. Nonetheless, the method demonstrates excellent compatibility with a wide range of functional groups, and products show strong potential for derivatization.

Question 2: Same time authors should have shown few good applications of synthesized chiral products for natural products synthesis or drugs synthesis.

Response: To highlight utility, we oxidized the aldehyde group of the products and condensed them with various bioactive molecules (tryptamine, rivaroxaban, fluvoxamine, oseltamivir, dehydroabietylamine, linagliptin, geraniol, cholesterol, diacetone-D-glucose, pregnenolone, L-menthol, and testosterone), generating biologically relevant BCH analogs in high yields.

Question 3: Same time authors can give some more supporting data for mechanism with DFT or/and kinetic data.

Response: We supplemented our mechanistic studies with DFT calculations. These confirm that trifluoroacetic acid lowers the reaction barrier by activating the BCB substrate, thereby validating our proposed mechanism.

Response to reviewer #1 :

Comments: *The authors have significantly improved the scientific quality of the manuscript. All the concerns have been satisfactorily addressed and I would like to suggest the acceptance at the current form.*

Response: We appreciate the time and effort you have dedicated to providing feedback on our manuscript. Thank you for your valuable suggestions, which have greatly contributed to improving the quality of our paper.

Response to reviewer #3 :

Comments: *Authors have attempted to give answers for reviewer comments raised in the revised manuscript of an amine-catalyzed enantioselective [2+2]-cycloaddition of BCBs with α,β -unsaturated aldehydes in moderate to poor yields with good to excellent ee's. The protocol operated under ambient conditions with more optimized conditions, demonstrating with good functional group tolerance while delivering products with good enantioselectivity (up to 99% ee). Furthermore, good number of transformations of the products were achieved, enabling access to derivatives. Controlled experiments confirmed an acid-promoted secondary amine catalysis pathway in the revised manuscript.*

This reviewer agree that authors reported a good platform for the synthesis of chiral BCHs and also provided a rare example of asymmetric BCB cycloaddition products. This reviewer believes that this revised article is mature enough to publish in Nature Communications as new protocol.

Response: We appreciate the time and effort you have dedicated to providing feedback on our manuscript. Thank you for your valuable suggestions, which have greatly contributed to improving the quality of our paper.